# SAMITE: Position Prompted SAM2 with Calibrated Memory for Visual Object Tracking

## Abstract

Visual Object Tracking (VOT) is widely used in applications like autonomous driving to continuously track targets in videos. Existing methods can be roughly categorized into template matching and autoregressive methods, where the former usually neglects the temporal dependencies across frames and the latter tends to get biased towards the object categories during training, showing weak generalizability to unseen classes. To address these issues, some methods propose to adapt the video foundation model SAM2 for VOT, where the tracking results of each frame would be encoded as memory for conditioning the rest of frames in an autoregressive manner. Nevertheless, existing methods fail to overcome the challenges of object occlusions and distractions, and do not have any measures to intercept the propagation of tracking errors. To tackle them, we present a SAMITE model, built upon SAM2 with additional modules, including: (1) Prototypical Memory Bank: We propose to quantify the feature-wise and position-wise correctness of each frame's tracking results, and select the best frames to condition subsequent frames. As the features of occluded and distracting objects are feature-wise and position-wise inaccurate, their scores would naturally be lower and thus can be filtered to intercept error propagation; (2) Positional Prompt Generator: To further reduce the impacts of distractors, we propose to generate positional mask prompts to provide explicit positional clues for the target, leading to more accurate tracking. Extensive experiments have been conducted on six benchmarks, showing the superiority of SAMITE. The code will be released upon paper acceptance.

## 1 Introduction

Visual Object Tracking (VOT) (Bertinetto et al., 2016; Chen et al., 2021; Li et al., 2019; Wu et al., 2013) constitutes a fundamental challenge in computer vision, aiming to continuously locate an arbitrary target in video sequences based on its initial state (e.g., bounding box). As a critical enabler for applications ranging from autonomous driving (Ettinger et al., 2021; Leon & Gavrilescu, 2021) to surveillance systems (Hsieh et al., 2006; Fuentes & Velastin, 2006), VOT requires robust mechanisms to handle dynamic scenarios where targets undergo appearance variations, occlusions, scale changes, and environmental distractions.

Mainstream VOT methods can be roughly divided into template matching methods (Ye et al., 2022; Cui et al., 2022; Gao et al., 2022; Hong et al., 2024) and autoregressive methods (Chen et al., 2023; Wei et al., 2023; Bai et al., 2024; Shi et al., 2024). The first type usually takes the first frame, with ground truth (GT), as the template to match the target objects in other search frames, *regardless of the temporal information*. Instead, autoregressive methods condition the current frame with the coordinates of predicted objects in previous frames to capture temporal dependencies. However, they may get biased towards the object categories appearing in training data (Zhou et al., 2024), showing *weak generalizability*.

To address this, a few recent advances (Yang et al., 2024; Videnovic et al., 2024) propose to leverage the robust video foundation model SAM2 (Ravi et al., 2024) for VOT. SAM2 is a promptable memory-based video segmentation model, where the predicted object in each frame would be encoded as memory, further used to condition subsequent frames. By default, the memories of the first prompted frame and 6 most recent frames are selected to predict 3 masks (with different confidences) for the current frame, where the one with the highest confidence is taken as the current

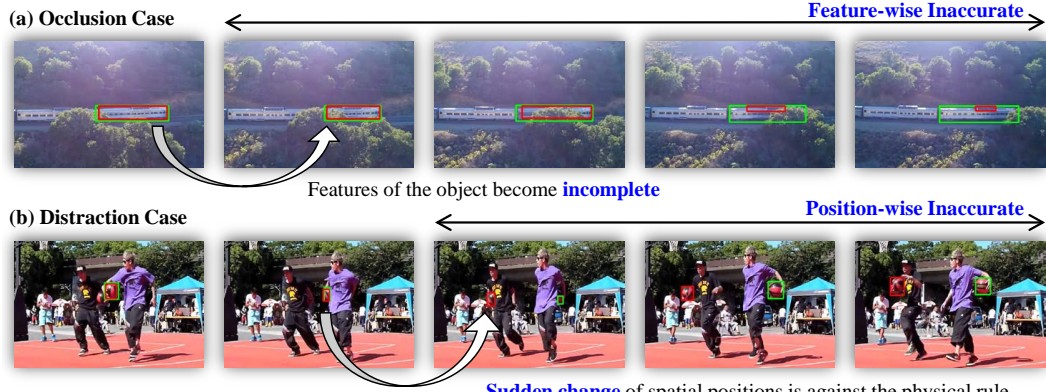

Figure 1: Two failure cases of *occlusion* and *distraction*. (a) The occluded frames shall not be selected to condition subsequent frames, as the tracking target may be reduced from the train carriage to louver; (b) The spatial positions of target objects in adjacent frames should be close.

prediction. Nevertheless, such mechanism cannot overcome the challenges of *occlusions* (Yang et al., 2024) and *distractions* (Videnovic et al., 2024). Therefore, SAMURAI (Yang et al., 2024) utilizes SAM2's intrinsic occlusion prediction head to estimate the likelihood of occlusion to filter memories and additionally maintains a Kalman Filter (Kalman, 1960) to preserve memories with good motions. In addition, SAM2.1++ (Videnovic et al., 2024) proposes to quantify the gaps between 3 predicted masks for each frame, where a large gap exhibits the existence of distracting objects, and this frame will be excluded from selection.

As shown in Figure 1, *occlusion* and *distraction* issues have not been well resolved, and existing methods (Yang et al., 2024; Videnovic et al., 2024) could hardly intercept *error propagation*, leading to long-term wrong tracking. In Figure 1(a), we observe that the occlusion likelihoods estimated by SAMURAI remain high when the target objects are occluded but not completely. As a result, memories with occlusions would still be used to condition subsequent frames, despite the fact that these target features are incomplete, i.e., *feature-wise inaccurate*, leading to ambiguous tracking targets, e.g., from train carriage to louver. In Figure 1(b), despite SAM2.1++ trying to resolve the distraction issue, the 3 masks generated for each frame usually locate the same object, i.e., the mechanism designed to detect distraction appears to be fragile. For example, the masks predicted for the middle frame uniformly locate the left basketball, so SAM2.1++ would not regard it as a distraction case, i.e., the memory of this frame would be used for subsequent frames, propagating the *position-wise inaccurate* information.

To alleviate the negative impacts of *occlusions* and *distractions*, we design a zero-shot **SAMITE** model, which is built upon SAM2 and includes: (1) **Prototypical Memory Bank**: We extend the memory banks (Ravi et al., 2024; Yang et al., 2024; Videnovic et al., 2024) by additionally extracting and storing a one-pixel prototype for each frame, representing the global descriptions of the predicted object. When selecting memories for the current frame, we first identify a **feature-wise anchor** (a frame with accurate prediction) and a **position-wise anchor** (a frame whose target is nearby to the current). Then, we regard other processed frames as candidates and measure 2 prototype-wise similarity scores between each candidate and the anchors. Finally, the frames with the best scores would be selected to condition the current frame. As a result, the errors raised by occlusions and distractions can be **intercepted** from long-term propagation; (2) **Positional Prompt Generator**: To further mitigate the distraction issue, we propose to use SAM2's promptable ability by generating **explicit positional clues** for the target object. Specifically, we follow AENet (Xu et al., 2024) to generate a prior mask to locate all objects with the same class as the target. After that, positional information is introduced to activate the target object while suppressing other distractors in the prior mask. Finally, the rectified prior mask is used as pseudo mask prompt to inform the model with the **position** of the target object, leading to more stable tracking.

To the best of our knowledge, we are the first to identify the *error propagation* issue raised by *occlusions* and *distractions*, when deploying SAM2 for VOT. To intercept the propagation, we design

a **Prototypical Memory Bank** to quantify the correctness of each predicted object, in terms of both feature and position, so as to excluding the errors from conditioning subsequent frames. Besides, we design **Positional Prompt Generator** to explicitly introduce some positional clues about the target object, mitigating the issue of distractions. Extensive experiments have been conducted on 6 benchmarks LaSOT, LaSOT$_{ext}$, GOT-10k, TrackingNet, NFS and OTB, demonstrating the superiority of the proposed methodology. Notably, our P$_{norm}$ score on LaSOT$_{ext}$ is 3.6%+ better than others.

## 2 RELATED WORK

**Visual Object Tracking (VOT).** Recent advancements primarily divide VOT methods into template matching (Bertinetto et al., 2016; Li et al., 2019; Chen et al., 2021; Yan et al., 2021a; Ye et al., 2022; Cui et al., 2022; Chen et al., 2020; Zhang et al., 2020; Guo et al., 2021; Xie et al., 2022; Danelljan et al., 2019; Yan et al., 2021b; Xu et al., 2020; Guo et al., 2020; Voigtlaender et al., 2020; Li et al., 2018; Mayer et al., 2021; Zhou et al., 2024) and autoregressive methods (Chen et al., 2023; Bai et al., 2024; Wei et al., 2023; Xie et al., 2024; Shi et al., 2024). Template matching methods mainly adopt a four-stage process: (1) The frame to be tracked is regarded as the search frame, and a template frame is offline matched for reference; (2) Extract features for two frames; (3) Match the features of the search frame with those of the template frame to activate the features of target; (4) Forward the activated features to the bounding box prediction head to obtain tracking results. However, the template-search matching belongs to one-to-one matching, making it challenging to capture temporal dependencies. Instead, autoregressive methods follow the sequential token prediction paradigm of language tasks, where the tracking results (e.g., the predicted coordinates) of the previous frame are utilized in the current frame in an autoregressive manner. In this way, the temporal dynamics can be well captured, while these methods show weak generalizability (Zhou et al., 2024) to unseen object categories (i.e., categories do not appear during training). Hence, we aim to leverage the well learned knowledge of video foundation models to address it.

**SAM2-based VOT.** Segment Anything Model (SAM) (Kirillov et al., 2023) has shown remarkable image segmentation ability, which supports prompts like points, bounding boxes and masks to segment relevant objects. Later, SAM2 (Ravi et al., 2024) has extended SAM by enabling promptable memory-based video object segmentation (VOS), which involves the selection of memories from tracked frames to condition the current frame, and has achieved good results in both VOS (Ding et al., 2024; Yang et al., 2025) and VOT (Yang et al., 2024; Videnovic et al., 2024). By default, the memories of the first prompted frames and 6 most recent frames are selected, regardless of challenges like object occlusions and distractions. To address them, some concurrent researches have introduced motion-based (Yang et al., 2024) and distractor-aware memory banks (Videnovic et al., 2024), with corresponding memory selection strategies. Nevertheless, the challenges have not been resolved, and once errors occur, these methods would fail to intercept error propagation, serving as the main motivation of this paper.

## 3 METHODOLOGY

### 3.1 REVISIT SAM2

SAM2 (Ravi et al., 2024) is a memory-based foundation model for segmenting objects in videos, which can naturally be adapted for VOT by simply converting the predicted binary masks to bounding boxes, and an illustration is provided in Appendix E.1. Some details are explained below.

**Memory Bank, Memory Attention and Memory Encoder.** After extracting the features of a frame, SAM2 would select the memories of the first frame (with GT) and 6 most recent frames from the Memory Bank, and use Memory Attention to condition the target object in features, which are decoded by Mask Decoder to obtain mask predictions. Memory Encoder will encode the features and the predicted mask as memory, representing the target object's information in current frame. This memory will be added to the Memory Bank for conditioning subsequent frames.

**Prompt Encoder and Mask Decoder.** SAM2 supports sparse (points, bounding boxes) and dense (masks) prompts to identify target objects. Prompt Encoder will represent the sparse prompts as the sum of both positional and learned encodings for each type, while the dense prompts are embedded by convolutions, further summed with image features to activate the target objects. The Mask De-

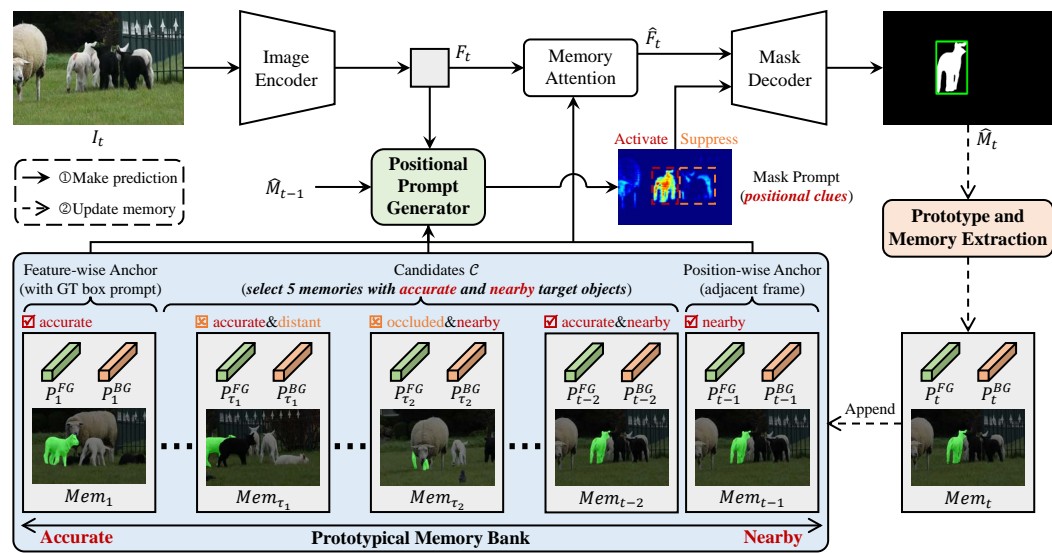

Figure 2: Overview of **SAMITE**, which is built upon SAM2, including: (1) **Prototypical Memory Bank (PMB)** is responsible for selecting calibrated memories with **accurate** and **nearby** target objects; (2) **Positional Prompt Generator (PPG)** generates pseudo mask prompt that can activates the target object, while suppressing other distracting objects, acting as **positional clues**.

coder deploys a two-way transformer to decode either *non-conditioned* yet **prompted** features (for the first frame) or the **memory-conditioned** yet *unprompted* features (for other frames). SAM2 uses multi-head branches (Yang et al., 2024) to generate multiple masks with corresponding confidences, where the one with the highest confidence will be the final prediction.

## 3.2 SAMITE

As shown in Figure 2, we build SAMITE upon SAM2 to tackle the aforementioned issues.

**Initialize Tracking with Prompt.** The first frame $I_1$ is forwarded to Image Encoder to extract its features $F_1$, which are directly decoded by Mask Decoder with GT bounding box prompt. The predicted mask $\hat{M}_1$ and image features $F_1$ are processed by Prototype and Memory Extraction module to obtain the corresponding foreground (FG) prototype $P_1^{FG}$, background (BG) prototype $P_1^{BG}$, and memory $Mem_1$, which are added to Prototypical Memory Bank (PMB) (Section 3.2.1).

**Memory-conditioned Tracking.** Any other frames $t$ is uniformly forwarded to Image Encoder to extract features $F_t$. Then, we select 7 memories and prototypes from PMB, containing accurate and nearby target. $F_t$ are fused with the selected memories as $\hat{F}_t$, activating the target object. Besides, Positional Prompt Generator (PPG) (Section 3.2.2) is deployed to generate positional mask prompt $\tilde{M}_t$, based on the selected prototypes. Next, features $\hat{F}_t$ and prompt $\tilde{M}_t$ are decoded into mask prediction $\hat{M}_t$. Finally, the FG and BG prototypes $P_t^{FG}$ and $P_t^{BG}$, as well as the memory $Mem_t$, of the current frames would be extracted and updated to PMB, used to condition subsequent frames.

### 3.2.1 PROTOTYPICAL MEMORY BANK (PMB)

The details of PMB are included in Figure 2, which are formally described as follows.

**Prototype and Memory Extraction.** After obtaining the mask prediction $\hat{M}_t \in \{0, 1\}^{H \times W \times 1}$ of current frame $t$, we first use SAM2's Memory Encoder to encode the current features $F_t \in \mathbb{R}^{H \times W \times C}$ into memory $Mem_t \in \mathbb{R}^{H \times W \times C}$, where $H$, $W$ and $C$ represent the height, width and hidden dimension of features. Then, we compress the features $F_t$ into FG and BG prototypes $P_t^{FG} \in \mathbb{R}^{1 \times C}$ and $P_t^{BG} \in \mathbb{R}^{1 \times C}$ via global average pooling, representing global descriptions of FG (target object) and BG (other objects), respectively. The obtained memory and prototypes will

be offloaded from GPU to CPU and updated to PMB. This procedure can be written as:

$$Mem_t = \text{MemoryEncoder}(F_t, \hat{M}_t) \tag{1}$$

$$P_t^{FG} = \text{GAP}(F_t, \hat{M}_t), P_t^{BG} = \text{GAP}(F_t, 1 - \hat{M}_t) \tag{2}$$

$$\text{PMB} = \text{AddMemory}(\text{PMB}, Mem_t, P_t^{FG}, P_t^{BG}) \tag{3}$$

**Memory Calibration.** To intercept *error propagation*, we propose to score and sort each frame, then select the most appropriate ones for robust memory conditioning. Since the first frame has been prompted by GT, its memory would be **accurate**. Besides, videos usually have 30 frames per second (FPS), the target object in last frame $t-1$ is naturally **nearby** to that in current frame $t$. Therefore, we regard their FG prototypes $P_1^{FG} \in \mathbb{R}^{1 \times C}$ (frame 1) and $P_{t-1}^{FG} \in \mathbb{R}^{1 \times C}$ (frame $t-1$) as the **feature-wise anchor** and the **position-wise anchor**, and their corresponding memories are uniformly taken as 2 out of 7 selected memories. For each of other previous frames $\tau$ ($2 \leq \tau \leq t-2$), 2 cosine similarities are measured between its FG prototype $P_\tau^{FG}$ and 2 anchors as follows.

$$S_\tau^{Feat} = \text{Reshape}(\text{Norm}(\text{Cos}(P_\tau^{FG}, P_1^{FG}))), S_\tau^{Pos} = \text{Reshape}(\text{Norm}(\text{Cos}(P_\tau^{FG}, P_{t-1}^{FG}))) \tag{4}$$

$$S_\tau = (1-\alpha) \cdot S_\tau^{Feat} + \alpha \cdot S_\tau^{Pos} \tag{5}$$

where $S_\tau^{Feat} \in [0,1]^1$ stands for the feature-wise score of frame $\tau$, $\text{Norm}(\cdot)$ means normalizing the similarity scores to value range $[0,1]$, $\text{Cos}(\cdot)$ is the cosine similarity operator, $S_\tau^{Pos} \in [0,1]^1$ is the position-wise score, and $S_\tau \in [0,1]^1$ is the final score re-weighted by hyperparameter $\alpha$ (empirically set to 0.3 in our experiments, please refer to Appendix D.4 for the parameter study of $\alpha$).

$$\text{Ind} = \text{Top5}(S_{\mathcal{C}}) \tag{6}$$

$$P^{FG}, P^{BG}, Mem = \text{SelectMemory}(\text{PMB}, \text{Ind}) \tag{7}$$

where Ind denotes the indices of selected frames, $\text{Top5}(\cdot)$ means sorting and taking the top 5 scores, $\mathcal{C}$ forms the set of candidate frames. The calibrated memories $Mem \in \mathbb{R}^{7 \times H \times W \times C}$, including 2 anchor and 5 selected frames' memories, are used in Memory Attention to condition features as $\hat{F}_t$, while the corresponding prototypes $P^{FG} \in \mathbb{R}^{7 \times C}$ and $P^{BG} \in \mathbb{R}^{7 \times C}$ are used in Section 3.2.2 to generate positional mask prompt. The visual impacts of this process are detailed in Appendix E.2.

**Reduced Candidate Set.** The candidate set $\mathcal{C}$ includes frame 2 to $t-2$ by default, so the computational cost of Eq. 4 would be quite large when dealing with long videos, e.g., the candidate set $\mathcal{C}$ of current frame 10001 will include 10,000 candidates and the cost of each cosine similarity is $10000 \times 1$. Fortunately, we observe it sufficient to only **consider recent frames** (e.g., frame $t-m$ to frame $t-2$) **as candidates**, the rationalities comprise: (1) When a video contains a moving target, the spatial positions of the target are likely to be different between earlier and more recent frames; (2) As the moving of objects is continuous and the positions of objects usually cannot be greatly changed within a short period, we empirically set $m = 30$ to select candidate frames, greatly reducing the computational complexity. The parameter study of $m$ is included in Appendix D.4.

**Computational Complexity.** The computational burden is mainly introduced by Eq. 4, where similarities are measured among prototypes. The cost is $\mathcal{O}(2m)$ for each frame, where $m = 30$ is the predefined candidate number, so the cost appears to be quite cheap.

### 3.2.2 POSITIONAL PROMPT GENERATOR (PPG)

As shown in Figure 3, PPG is designed to generate position-enhanced prior masks, further used as pseudo mask prompt to distinguish the target object from distracting object (e.g., other visually-similar birds in the figure). This module is formally described as follows.

**Positional Prior Mask.** Inspired by prior masks (Tian et al., 2020; Xu et al., 2024) that can roughly locate the same-class objects in one image, we follow AENet (Xu et al., 2024) to generate discriminative prior masks, and introduce positional information to rectify them. Firstly, cosine similarities are measured between current frame's features $F_t \in \mathbb{R}^{H \times W \times C}$ and FG and BG prototypes $P_\tau^{FG}$ and $P_\tau^{BG}$, where $\tau$ denotes each of the 7 selected frames (in Section 3.2.1).

$$\tilde{M}_\tau^{FG} = \text{Reshape}(\text{Norm}(\text{Cos}(F_t, P_\tau^{FG}))), \tilde{M}_\tau^{BG} = \text{Reshape}(\text{Norm}(\text{Cos}(F_t, P_\tau^{BG}))) \tag{8}$$

$$\tilde{M}_\tau^{Disc} = \text{Norm}(\text{ReLU}(\tilde{M}_\tau^{FG} - \tilde{M}_\tau^{BG})) \tag{9}$$

Figure 3: Details about **Positional Prompt Generator (PPG)**. Take bird-15 of LaSOT as an example, where frame #501 is the current frame, and frame #471 is one of the selected frames.

where $\tilde{M}_\tau^{FG} \in [0,1]^{H \times W \times 1}$ and $\tilde{M}_\tau^{BG} \in [0,1]^{H \times W \times 1}$ show the probability of each pixel being considered as FG (target objects) or BG (other objects), respectively. $\tilde{M}_\tau^{Disc} \in [0,1]^{H \times W \times 1}$ means whether a pixel is more likely to be FG rather than BG.

As we can observe from the figure, $\tilde{M}^{Disc}$ can **effectively locate the same-class objects**, yet *more than just the target*. Therefore, we aim to introduce extra positional information to **preserve the target object** and **suppress distracting objects**. Specifically, we generate 2D positional encodings (Wang & Liu, 2019) $E_{pos} \in [0,1]^{H \times W \times C}$ for features $F_t$. As the target object in current frame $t$ is naturally close to that in last frame $t-1$, we take mask prediction $\hat{M}_{t-1}$ to obtain the positional prototype $P_{t-1}^{Pos} \in \mathbb{R}^{1 \times C}$, further utilized to generate positional prior mask $\tilde{M}_{t-1}^{Pos}$ as follows.

$$E^{Pos} = \text{PE}(F_t), P_{t-1}^{Pos} = \text{GAP}(E^{Pos}, \hat{M}_{t-1}) \tag{10}$$

$$\tilde{M}_{t-1}^{Pos} = \text{Reshape}(\text{Norm}(\text{Cos}(E^{Pos}, P_{t-1}^{Pos}))) \tag{11}$$

We can observe that the target object is less likely to appear in distant positions. Finally, we use the positional prior mask to re-weight the discriminative prior mask, which is then normalized to be the positional mask prompt $\tilde{M}_\tau \in \mathbb{R}^{H \times W \times 1}$, generated by previous frame $\tau$. Kindly note that each of the selected frames will generate a positional mask prompt, and we simply average them to be the final prompt for current frame. Some visualizations are included in Section 4.3 and Appendix E.3.

$$\tilde{M}_\tau = \text{Norm}(\tilde{M}_\tau^{Disc} \cdot \tilde{M}_{t-1}^{Pos}) \tag{12}$$

$$\tilde{M}_t = \text{Avg}(\tilde{M}_\tau), \tau \in \mathcal{C} \tag{13}$$

The generated pseudo mask prompt is then encoded by Memory Encoder and forwarded to Mask Decoder to segment the memory-conditioned features $\hat{F}_t$ as $\hat{M}_t$.

**Cycle-Consistent Checking.** As mask prompts are generated solely based on similarities, they are never as accurate as the unavailable GTs. We observe in cases like severe motion blurs, the prompts are likely to be inaccurate. Therefore, we further presents a checking mechanism to determine if the generated prompt should be used for current frame or not. Thanks to the lightweight computations of Mask Decoder, we expand the batch dimension of the memory-conditioned features, and parallelly decode them into 2 mask predictions (with and without prompt). Then, we generate prototypes for the current frame, and use Eq. 8 to Eq. 12 to **reversely generate prompts for each selected memory**. Finally, they are binarized and measured mean intersection-over-union (mIoU) with the mask prediction of each memory. Each frame would be prompted, only if the averaged mIoU exceeds $\beta$. The parameter study of $\beta$ is included in Appendix D.4.

**Computational Complexity.** The complexity is raised by Eq. 8 and Eq. 11, where the former would be repeated 7 times (7 memories), and the latter is conducted only once. They will be used in both Positional Prior Mask and Cycle-Consistent Checking, therefore, the overall cost is $\mathcal{O}(2 \times (7+1) \times H \times W) = 16HW$ for each frame, where $16 \ll HW$, exhibiting linear complexity.

Table 1: Quantitative comparisons with state-of-the-arts on LaSOT, $\text{LaSOT}_{ext}$ and GOT-10k. "#Param" is the parameter number in million. The methods are sorted based on AUC of LaSOT. **Bold** values denote the best performance. Underlined results indicate the second best.

| Trackers | Source | #Param | LaSOT | | | $\text{LaSOT}_{ext}$ | | | GOT-10k | | |
|---|---|---|---|---|---|---|---|---|---|---|---|
| | | | AUC | $P_{norm}$ | P | AUC | $P_{norm}$ | P | AO | $SR_{0.5}$ | $SR_{0.75}$ |
| *Supervised VOT methods* | | | | | | | | | | | |
| DiMP-50$_{288}$ (Bhat et al., 2019) | ICCV'19 | 46 | 56.9 | 65.0 | - | - | - | - | 61.1 | 71.7 | 49.2 |
| AutoMatch$_{255}$ (Zhang et al., 2021) | ICCV'21 | 24 | 58.2 | 67.5 | 59.9 | 37.6 | - | 43.0 | 65.2 | 76.6 | 54.3 |
| PrDiMP50$_{288}$ (Danelljan et al., 2020) | CVPR'20 | 43 | 59.8 | 68.8 | - | - | - | - | 63.4 | 73.8 | 54.3 |
| TransT$_{256}$ (Chen et al., 2021) | CVPR'21 | 23 | 64.9 | 73.8 | 69.0 | - | - | - | 67.1 | 76.8 | 60.9 |
| STARK-101$_{320}$ (Yan et al., 2021a) | ICCV'21 | 47 | 67.1 | 76.9 | 72.2 | - | - | - | 68.8 | 78.1 | 64.1 |
| AiATrack (Gao et al., 2022) | ECCV'22 | 18 | 69.0 | 79.4 | 73.8 | 47.7 | 55.6 | 55.4 | 69.6 | 80.0 | 63.2 |
| MixFormer$_{320}$ (Cui et al., 2022) | CVPR'22 | 37 | 69.2 | 78.7 | 74.7 | - | - | - | - | - | - |
| GRM$_{256}$ (Gao et al., 2023) | CVPR'23 | 100 | 69.9 | 79.3 | 75.8 | - | - | - | 73.4 | 82.9 | 70.4 |
| OSTrack$_{384}$ (Ye et al., 2022) | ECCV'22 | 93 | 71.1 | 81.1 | 77.6 | 50.5 | 61.3 | 57.6 | 73.7 | 83.2 | 70.8 |
| SwinTrack-B$_{384}$ (Lin et al., 2022) | NIPS'22 | 91 | 71.3 | - | 76.5 | 49.1 | - | 55.6 | 72.4 | 80.5 | 67.8 |
| ROMTrack$_{384}$ (Cai et al., 2023) | ICCV'23 | 92 | 71.4 | 81.4 | 78.2 | 51.3 | 62.4 | 58.6 | 74.2 | 84.3 | 72.4 |
| SeqTrack-B$_{384}$ (Chen et al., 2023) | CVPR'23 | 89 | 71.5 | 81.1 | 77.8 | 50.5 | 61.6 | 57.5 | 74.5 | 84.3 | 71.4 |
| LoRAT-B$_{378}$ (Lin et al., 2024) | ECCV'24 | 99 | 72.4 | 81.8 | 79.1 | 52.9 | 64.5 | 60.6 | 73.7 | 82.6 | 72.9 |
| EVPTrack$_{384}$ (Shi et al., 2024) | AAAI'24 | 73 | 72.7 | 82.9 | 80.3 | 53.7 | 65.5 | 61.9 | 76.6 | 86.7 | 73.9 |
| HIPTrack$_{384}$ (Cai et al., 2024) | CVPR'24 | 120 | 72.7 | 82.9 | 79.5 | 53.0 | 64.3 | 60.6 | 77.4 | 88.0 | 74.5 |
| AQATrack$_{384}$ (Xie et al., 2024) | CVPR'24 | 72 | 72.7 | 82.9 | 80.2 | 52.7 | 64.2 | 60.8 | 76.0 | 85.2 | 74.9 |
| ARTrackV2$_{384}$ (Bai et al., 2024) | CVPR'24 | 135 | 73.0 | 82.0 | 79.6 | 52.9 | 63.4 | 59.1 | 77.5 | 86.0 | **75.5** |
| ODTrack-B$_{384}$ (Zheng et al., 2024) | AAAI'24 | 93 | 73.2 | 83.2 | 80.6 | 52.4 | 63.9 | 60.1 | 77.0 | 87.9 | 75.1 |
| *Zero-shot SAM2-based methods* | | | | | | | | | | | |
| SAM2.1-B (Ravi et al., 2024) | ICLR'25 | 81 | 66.0 | 73.5 | 71.0 | 55.5 | 67.2 | 64.6 | 77.9 | 88.6 | 71.5 |
| SAMURAI-B (Yang et al., 2024) | Arxiv'24 | 81 | 70.7 | 78.7 | 76.2 | 57.5 | 69.3 | 67.1 | **79.6** | **90.8** | 72.9 |
| SAM2.1++-B (Videnovic et al., 2024) | CVPR'25 | 81 | 72.9 | 81.0 | 78.7 | 58.5 | 69.5 | 69.0 | 78.1 | 88.5 | 70.9 |
| SAMITE-B | Ours | 81 | **74.9** | **83.4** | **81.4** | **60.7** | **73.1** | **71.2** | 78.9 | 89.9 | 72.5 |

## 4 EXPERIMENTS

### 4.1 EXPERIMENTAL SETUP

**Datasets.** We follow SAMURAI (Yang et al., 2024) to verify the zero-shot performance of SAMITE on 6 benchmarks, including **LaSOT** (Fan et al., 2019) (280 videos, 2448 frames in average, 30 FPS), **LaSOT**$_{ext}$ (Fan et al., 2021) (150 videos, 2393 frames in average, 30 FPS), **GOT-10k** (Huang et al., 2019) (180 videos, 126 frames in average, 10 FPS), **TrackingNet** (Muller et al., 2018) (511 videos, 441 frames in average, 30 FPS), **NFS** (Kiani Galoogahi et al., 2017) (100 videos, 480 frames in average, 30 FPS) and **OTB** (Wu et al., 2015) (100 videos, 598 frames in average, 30 FPS). These datasets contain videos with various categories, different length for long or short-term tracking, and diverse challenges like occlusions or fast-moving objects, facilitating comprehensive evaluation for VOT. Please refer to Appendix C for more details about these datasets.

**Evaluation Metrics.** Following existing methods (Yang et al., 2024), Area Under the Curve (AUC), Normalized Precision ($P_{norm}$) and Precision (P) are deployed to evaluate the performance on LaSOT, $\text{LaSOT}_{ext}$. Average Overlap (AO) and Success Rate ($SR_{0.5}$ and $SR_{0.75}$) are used to evaluate GOT-10k. Only AUC is utilized to evaluate TrackingNet, NFS and OTB.

### 4.2 COMPARISONS WITH STATE-OF-THE-ARTS

**Quantitative Results.** The quantitative comparisons between SAMITE and state-of-the-arts are presented in Table 1 (for LaSOT, $\text{LaSOT}_{ext}$ and GOT-10k) and Table 2 (for TrackingNet, NFS and OTB), and we would like to make the following notes: (1) Since VOT is a task requiring real-time inference, we select the best version of each method whose parameter number does not exceed 150 million for fair comparisons; (2) The methods can be roughly divided into 2 categories, including supervised VOT methods and zero-shot SAM2-based methods, where the latter uniformly deploys robust SAM2, trained on the video object segmentation (VOS) (Caelles et al., 2017) task, to directly perform testing for VOT; (3) $\text{LaSOT}_{ext}$ and GOT-10k are 2 challenging datasets where the training and testing object categories do not overlap, i.e., the case is out-domain testing.

We can observe from the tables that: (1) Though SAMITE has not been trained on VOT datasets, it can achieve the best performance in most situations, e.g., the AUC score on LaSOT is 1.7% and 2% better than the best VOT method ODTrack (74.9% v.s. 73.2%) and SAM2-based method

Table 2: Quantitative comparisons with state-of-the-arts on TrackingNet (TN), NFS and OTB.

| Trackers | Source | TN | NFS | OTB |
|---|---|---|---|---|
| *Supervised VOT methods* | | | | |
| AiATrack (Gao et al., 2022) | ECCV'22 | 82.7 | 67.9 | 69.6 |
| OSTrack$_{384}$ (Ye et al., 2022) | ECCV'22 | 83.9 | 66.5 | 55.9 |
| SeqTrack-B$_{384}$ (Chen et al., 2023) | CVPR'23 | 83.9 | 66.7 | - |
| GRM$_{256}$ (Gao et al., 2023) | CVPR'23 | 84.0 | 65.6 | - |
| ROMTrack$_{384}$ (Cai et al., 2023) | ICCV'23 | 84.1 | 68.8 | 70.9 |
| LoRAT-B$_{378}$ (Lin et al., 2024) | ECCV'24 | 84.2 | 68.3 | 70.9 |
| HIPTrack$_{384}$ (Cai et al., 2024) | CVPR'24 | 84.5 | 68.1 | **71.0** |
| *Zero-shot SAM2-based methods* | | | | |
| SAM2.1-B (Ravi et al., 2024) | ICLR'25 | 80.7 | 69.0 | 59.9 |
| SAMITE-B | Ours | **84.5** | **69.2** | 69.9 |

Table 3: Component-wise Ablation Study. "$\mathcal{A}$" refer to 2 anchors. "PPM" is Positional Prior Mask, and "CCC" is Cycle Consistent Checking.

| PMB | | PPG | | LaSOT | | |
|---|---|---|---|---|---|---|
| $\mathcal{A}_{Feat}$ | $\mathcal{A}_{Pos}$ | PPM | CCC | AUC | $P_{norm}$ | P |
| | | | | 72.1 | 80.1 | 77.8 |
| ✓ | | | | 73.7 | 82.0 | 79.9 |
| | ✓ | | | 73.7 | 81.9 | 79.7 |
| ✓ | ✓ | | | 74.6 | 82.9 | 80.9 |
| | | ✓ | | 71.9 | 78.5 | 76.8 |
| | | ✓ | ✓ | 73.2 | 81.6 | 79.2 |
| ✓ | ✓ | ✓ | ✓ | **74.9** | **83.4** | **81.4** |

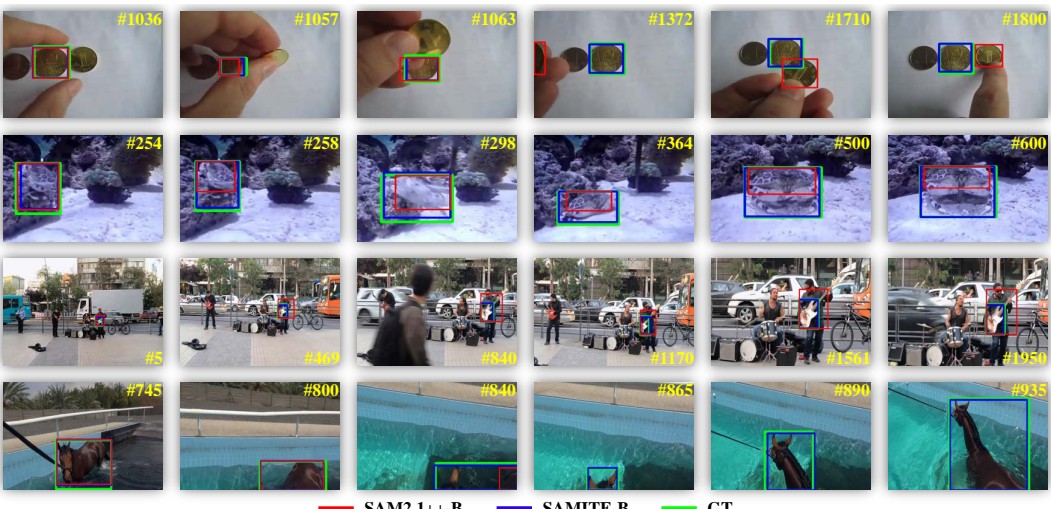

Figure 4: Qualitative comparisons with SAM2.1++ (Videnovic et al., 2024).

SAM2.1++ (74.9% v.s. 72.9%), respectively. Notably, for LaSOT$_{ext}$ dataset, our P$_{norm}$ score is 7.6% and 3.6% better than EVPTrack (73.1% v.s. 65.5%) and SAM2.1++ (73.1% v.s. 69.5%), **demonstrating the effectiveness of our module designs**; (2) In LaSOT$_{ext}$ and GOT-10k datasets where the testing object categories are unseen during training, there exists a prominent performance gap between VOT and SAM2-based methods, **validating the rationality of introducing SAM2 for VOT** (in Section 1). For example, their best AUC scores are 53.7% and 60.7% in LaSOT$_{ext}$.

**Qualitative Results.** We visually compare SAMITE with one of the best baselines SAM2.1++ (Videnovic et al., 2024), and present 4 examples in Figure 4: (1) A hand picks up the rightmost coin (distractor) and moves to the left, during which the middle coin (target) is occluded for a while, and SAM2.1++ fails to continuously focus on the target; (2) The crab looks similar to the surroundings (distractor), thus, SAM2.1++ mistakenly considers its legs as non-target at an early stage; (3) Our SAMITE is more discriminative than SAM2.1++, as it can distinguish different parts of the guitar in the third row; (4) When the horse moves outside the screen and then re-enters, SAM2.1++ fails to re-recognize the horse as the target. In brief, SAMITE is more capable of dealing with occlusions and distractors. More visualizations are included in Appendix E.4.

## 4.3 ABLATION STUDY

**Component-wise Ablation Study.** The detailed component-wise ablation study is presented in Table 3 to validate the effectiveness of module designs. We start with a tailored model, without PMB and PPG, and the initial AUC is 72.1%. When we merely include PMB and use either feature-wise anchor or position-wise anchor to sort and select the memories of processed frames, the AUC

scores are uniformly 73.7%, showing the necessity of memory calibration. If we use both anchors, the AUC can reach 74.6%, already surpassing all baselines in Table 1. The visual impacts of PMB are illustrated in Appendix E.2. For PPG, if we input the generated PPM as pseudo mask prompt for all frames, the AUC score is decreased from 72.1% to 71.9%, and we attribute it to the fact that the generated mask prompts are never as accurate as the unavailable GT masks, and may introduce some noises to deteriorate the performance. Hence, we further design CCC to post-check the generated prompts for selectively introducing them to some of the frames, leading to a performance gain of 1.3%. Finally, when both PMB and PPG are deployed, the AUC, $P_{norm}$ and P scores can reach 74.9%, 83.4% and 81.4%, showing the superiority of SAMITE.

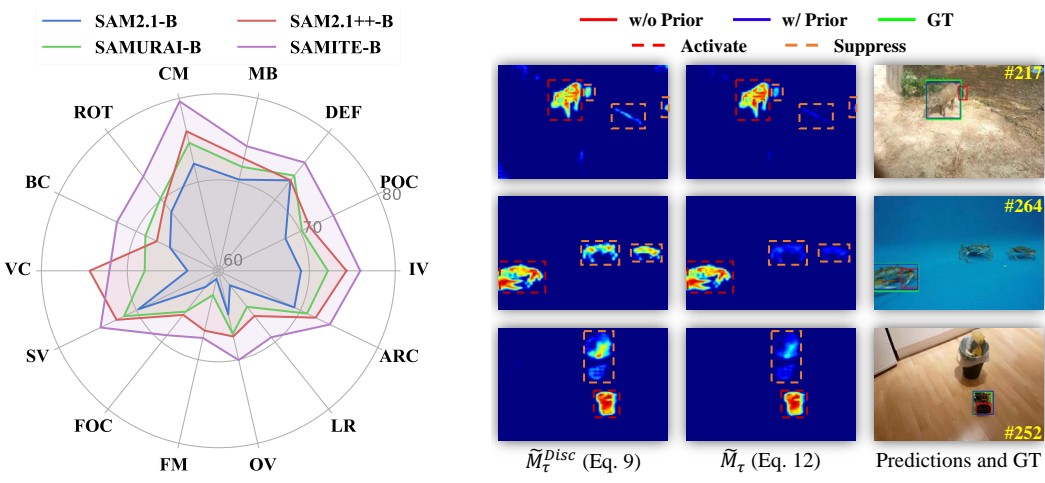

Figure 5: Attribute-wise Performance.        Figure 6: Visual impacts of PPG.

**Attribute-wise Performance on LaSOT.** LaSOT (Fan et al., 2019) defines 14 attributes (i.e., challenges) such as Full Occlusion for VOT, and annotates each sequence to indicate if each attribute exists or not. Following (Yang et al., 2024; Zheng et al., 2024; Bai et al., 2024; Cai et al., 2024), we compare the attribute-wise results in Figure 5. Kindly remind that the explanation for each abbreviation and the detailed values are included in Appendix D.1. It can be observed that our SAMITE is better at dealing with the challenges of VOT than other SAM2-based methods, e.g., for Full Occlusion (FOC) and Partial Occlusion (POC) scenarios, the AUC of SAMITE is 3.1% and 3.0% better than the best baseline SAM2.1++. Notably, SAMITE achieves 3.7% higher AUC than SAM2.1++, in terms of Background Clutter (BC) that denotes the existence of nearby distractors, **validating the claim of better at handling occlusions and distractions** in Section 1.

**Visual Impacts of Positional Prompt Generator.** We visualize 3 examples to show the visual impacts of PPG in Figure 3. The first and second columns refer to the prior masks without and with positional information. Note that the frames in the third column indicate where the predictions of 2 methods (without and with PPG) start to differ. We can observe that (1) the distractors are feature-wise similar to the target; (2) the generated positional prior mask can effectively suppress the distractors; and (3) the use of mask prompts can also help to ensure the completeness of objects, e.g., in the second and third rows, the tracked regions start to shrink if we do not use PPG.

## 5 CONCLUSION

In this paper, we present zero-shot SAMITE for Visual Object Tracking. SAMITE is built upon SAM2, with additional Prototypical Memory Bank (PMB) and Positional Prompt Generator (PPG) to alleviate the *error propagation* issue introduced by object *occlusions* and *distractions*. PMB aims to quantify the feature-wise and position-wise correctness of the tracking results of each processed frame, where the accurate ones would be selected to condition subsequent frames, so as to intercepting the propagation of errors. Besides, PPG is responsible for generating positional mask prompt, regarded as positional clues for distinguishing the target object from distractors. Extensive experiments have been conducted on 6 benchmarks, demonstrating the superiority of SAMITE.

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

## A    CODE

The source code is provided in the supplementary material. The instructions for reproducing our results are detailed in the README file. The code will be made public after paper acceptance.

## B    IMPLEMENTATION DETAILS

Experiments are conducted with single NVIDIA V100 with 32GB memory. SAMITE is built upon pretrained SAM2 (Ravi et al., 2024), and we follow SAMURAI (Yang et al., 2024) to evaluate its zero-shot performance on various VOT benchmarks. For the hyperparameters used in our designed modules, we set $\alpha = 0.3$ in Eq. 5 and $m = 30$ as the size of reduced candidate set for PMB, and set $\beta = 0.7$ as the threshold to use positional mask prompt in PPG, the parameter studies are included in Appendix D.4.. SAM2 has 4 versions with different sizes, including T (39M), S (46M), B (81M) and L (224M), after making trade-offs between performance and efficiency, we deploy SAM2-B (81M) in most of the experiments, and study the impacts of different model sizes in Appendix D.2.

## C    DETAILS OF DATASETS

The FPS of all datasets except GOT-10k (Huang et al., 2019) is 30 FPS, while GOT-10k's FPS is 10.

**LaSOT (Fan et al., 2019).** LaSOT dataset serves as a large-scale long-term tracking benchmark with 1,400 videos spanning 70 object categories, featuring an average sequence length of 2,448 frames. Its training and testing splits maintain a balanced distribution, containing 1,120 and 280 sequences, respectively, with 16 training and 4 testing samples per category.

**LaSOT$_{ext}$ (Fan et al., 2021).** LaSOT$_{ext}$ augments the original LaSOT dataset with 150 additional sequences across 15 novel object categories, featuring an average sequence length of 2,393 frames. These sequences are specifically curated to emphasize occlusion patterns and small-target variations, posing heightened challenges. Following established protocols, models trained exclusively on LaSOT (Fan et al., 2019) are evaluated in a zero-shot manner on this extension.

**GOT-10k (Huang et al., 2019).** It contains over 10,000 video segments depicting real-world moving objects, covering 560+ object classes and 80 distinct motion types. It adheres to a strict one-shot evaluation protocol, requiring trackers to be trained solely on domain-restricted training data, with 180 held-out videos for testing, prohibiting fine-tuning on test-domain information. GOT-10k is used for short-term tracking, and the average sequence length is 126 frames.

**TrackingNet (Muller et al., 2018).** This dataset is designed for short-term tracking (with an average video length of 441 frames), which offers a comprehensive dataset capturing diverse object classes in unconstrained environments, comprising 30,643 videos. Its split allocates 30,132 sequences for training and 511 for testing, ensuring generalizability across varied contextual scenarios.

**NFS (Kiani Galoogahi et al., 2017).** NFS includes 100 high-frame-rate (240 FPS) videos totaling 380,000 frames, simulating real-world motion dynamics. For compatibility with standard VOT frameworks, the 30 FPS subset with synthetically added motion blur is utilized, aligning with established practices in VOT research. For the 30 FPS version, the average frame number is 480.

**OTB (Wu et al., 2015).** OTB dataset represents one of the pioneering benchmarks for visual tracking, featuring 100 sequences (with an average length of 598 frames) systematically annotated with per-sequence attribute labels (e.g., occlusion, scale variation). This dataset facilitates fine-grained performance analysis across diverse tracking challenges.

## D    ADDITIONAL EXPERIMENTS

In this section, some additional experiments are provided, including details for attribute-wise performance on LaSOT (Appendix D.1), comparisons between SAM2-based methods with different parameter number (Appendix D.2), efficiency analysis (Appendix D.3), and parameter studies on hyperparameters (Appendix D.4).

## D.1 Detailed Attribute-wise Performance on LaSOT

Table 4: Details of the defined attributes in LaSOT.

| Attribute | Abbr. | Num | Definition |
|---|---|---|---|
| Camera Motion | CM | 86 | Abrupt motion of the camera |
| View Change | VC | 33 | Viewpoint affects target appearance significantly |
| Rotation | ROT | 175 | The target object rotates in the image |
| Scale Variation | SV | 273 | The ratio of target bounding box is outside the range [0.5, 2] |
| Deformation | DEF | 142 | The target object is deformable during tracking |
| Background Clutter | BC | 100 | The background near the target object has the similar appearance as the target |
| Partial Occlusion | POC | 187 | The target object is partially occluded in the sequence |
| Full Occlusion | FOC | 118 | The target object is fully occluded in the sequence |
| Motion Blur | MB | 89 | The target region is blurred due to the motion of target object or camera |
| Illumination Variation | IV | 47 | The illumination in the target region changes |
| Aspect Ratio Change | ARC | 249 | The ratio of bounding box aspect ratio is outside the rage [0.5, 2] |
| Out-of-View | OV | 104 | The target object completely leaves the video frame |
| Low Resolution | LR | 141 | The area of target box is smaller than 1000 pixels in at least one frame |
| Fast Motion | FM | 53 | The motion of target object is larger than the size of its bounding box |

Table 5: Attribute-wise AUC on LaSOT.

| Trackers | IV | POC | DEF | MB | CM | ROT | BC | VC | SV | FOC | FM | OV | LR | ARC |
|---|---|---|---|---|---|---|---|---|---|---|---|---|---|---|
| SAM2.1-B | 69.0 | 68.1 | 72.9 | 70.3 | 72.2 | 68.2 | 65.7 | 63.0 | 69.8 | 61.8 | 60.4 | 64.6 | 61.6 | 69.2 |
| SAMURAI-B | 72.2 | 70.2 | _73.6_ | 71.8 | 74.7 | _70.2_ | _68.8_ | 68.0 | 71.6 | 65.5 | 62.3 | 67.0 | 64.8 | 70.9 |
| SAM2.1++-B | _74.3_ | _71.2_ | 72.9 | _72.8_ | _76.1_ | 69.5 | 67.4 | **74.4** | _72.6_ | 66.0 | _66.6_ | _67.2_ | _66.1_ | _72.0_ |
| SAMITE-B | **75.9** | **74.3** | **75.5** | **74.3** | **79.6** | **73.4** | **72.5** | _72.1_ | **74.6** | **69.0** | **67.4** | **70.1** | **69.3** | **73.8** |
| Difference | 1.6 | 3.1 | 1.9 | 1.5 | 3.5 | 3.2 | 3.7 | -2.3 | 2.0 | 3.0 | 0.8 | 2.9 | 3.2 | 1.8 |

LaSOT (Fan et al., 2019) dataset defines 14 attributes, including Camera Motion (CM), View Change (VC), Rotation (ROT), Scale Variation (SV), Deformation (DEF), Background Clutter (BC), Partial Occlusion (POC), Full Occlusion (FOC), Motion Blur (MB), Illumination Variation (IV), Aspect Ratio Change (ARC), Out-of-View (OV), Low Resolution (LR) and Fast Motion (FM), for VOT. Their details are shown in Table 4.

LaSOT's test set comprises 280 videos, and each of them are carefully annotated based on whether each attribute exists or not. Benefiting from this characteristic, we compare the AUC scores of SAM2.1 (Ravi et al., 2024), SAMURAI (Yang et al., 2024), SAM2.1++ (Videnovic et al., 2024) and the proposed SAMITE model, and present the detailed results in Table 5. Compared with baselines, our SAMITE is particularly good at dealing with cases of Partial Occlusion (POC), Camera Motion (CM), Rotation (ROT), Background Clutter (BC), Full Occlusion (FOC), and Low Resolution (LR), where our AUC score appears to be 3.0%+ better than the second best. Notably, POC and FOC are related to object occlusions, and BC is exactly the distracting cases, so the improvements on these attributes validate the motivations and the effectiveness of our proposed methodology.

## D.2 SAM2-based Methods with Different Sizes

SAM2 has 4 variants in terms of different model sizes, including SAM2-T (39M), SAM2-S (46M), SAM2-B (81M) and SAM2-L (224M). To further study the impacts of different model sizes for SAM2-based methods, we conduct detailed analysis on the original SAM2.1 (Ravi et al., 2024), SAMURAI (Yang et al., 2024) and our proposed SAMITE. The experiments are conducted on La-SOT (Fan et al., 2019) and LaSOT$_{ext}$ (Fan et al., 2021), and the results are included in Table 6. The following conclusions can be drawn from the table: (1) SAMITE can consistently improve SAM2.1 by large margins, particularly, the AUC score of SAMITE-B can outperform that of SAM2.1 by 8.9%; (2) Both SAMURAI and SAMITE are built upon SAM2.1, SAMITE consistently performs better than SAMURAI, e.g., the gap of precision P can reach 5.2% when SAM2.1-B is deployed, showing the superiority of our module designs; (3) SAMITE can achieve more remarkable improvement with relative smaller model sizes include T, S and B, making it a wonderful solution for real-time inference in diverse real-world application scenarios like navigation.

Table 6: SAM2-based methods with different sizes.

| Size | #Param | Trackers | LaSOT | | | LaSOT$_{ext}$ | | |
|------|--------|----------|-----|-----------|-----|-----|-----------|-----|
| | | | AUC | P$_{norm}$ | P | AUC | P$_{norm}$ | P |
| T | 39 | SAM2.1 | 66.7 | 73.7 | 71.2 | 52.3 | 62.0 | 60.3 |
| | | SAMURAI | 69.3 | 76.4 | 73.8 | 55.1 | 65.6 | 63.7 |
| | | SAMITE | 72.8 | 80.6 | 78.3 | 57.5 | 68.0 | 66.2 |
| | | Difference | 3.5 | 4.2 | 4.5 | 2.4 | 2.4 | 2.5 |
| S | 46 | SAM2.1 | 66.5 | 73.7 | 71.3 | 56.1 | 67.6 | 65.8 |
| | | SAMURAI | 70.0 | 77.6 | 75.2 | 58.0 | 69.6 | 67.7 |
| | | SAMITE | 73.0 | 81.3 | 79.2 | 59.8 | 71.7 | 70.1 |
| | | Difference | 3.0 | 3.7 | 4.0 | 1.8 | 2.1 | 2.4 |
| B | 81 | SAM2.1 | 66.0 | 73.5 | 71.0 | 55.5 | 67.2 | 64.6 |
| | | SAMURAI | 70.7 | 78.7 | 76.2 | 57.5 | 69.3 | 67.1 |
| | | SAMITE | 74.9 | 83.4 | 81.4 | 60.7 | 73.1 | 71.2 |
| | | Difference | 4.2 | 4.7 | 5.2 | 3.2 | 3.8 | 4.1 |
| L | 224 | SAM2.1 | 68.5 | 76.2 | 73.6 | 58.6 | 71.1 | 68.8 |
| | | SAMURAI | 74.2 | 82.7 | 80.2 | 61.0 | 73.9 | 72.2 |
| | | SAMITE | 74.7 | 83.3 | 81.1 | 62.1 | 75.2 | 73.5 |
| | | Difference | 0.5 | 0.6 | 0.9 | 1.1 | 1.3 | 1.3 |

Table 7: Efficiency analysis. The unit of reported values is frame-per-second (FPS). All of SAM2.1, SAMURAI, SAM2.1++ and SAMITE are SAM2-based zero-shot methods, and the spatial resolution of video frames is $1024 \times 1024$.

| Size | #Param | SAM2.1 | SAMURAI | SAM2.1++ | SAMITE |
|------|--------|--------|---------|----------|--------|
| T | 39 | 17.3 | 17.3 | 11.3 | 10.9 |
| S | 46 | 17.3 | 17.3 | 11.2 | 10.9 |
| B | 81 | 14.5 | 14.3 | 9.2 | 9.2 |
| L | 224 | 10.0 | 10.0 | 6.1 | 7.8 |

### D.3 EFFICIENCY ANALYSIS

We conduct efficiency analysis on SAM2.1 (Ravi et al., 2024), SAMURAI (Yang et al., 2024), SAM2.1++ (Videnovic et al., 2024) and our SAMITE, and report their frame-per-second (FPS) in Table 7. All of them are SAM2-based zero-shot methods, uniformly adopt $1024 \times 1024$ video frames as input, and would select 7 memories (if possible) to condition each frame. We can observe from the table that: (1) SAMITE shows similar efficiency as the concurrent work SAM2.1++, but can achieve better results (as shown in Table 1); (2) With the increase of parameter number, the efficiency gap between SAMITE and SAM2.1 (i.e., the original SAM2 model) gets smaller, e.g., when using SAM2-B (46M), the gap in FPS is around 5; (3) Although the designed modules would make the inference speed slower, it is worthy since sacrificing 5 FPS (SAM2.1-B to SAMITE-B) can achieve an AUC gain of 8.9% on LaSOT.

### D.4 PARAMETER STUDIES ON HYPERPARAMETERS

In this section, we conduct experiments to study the impacts of hyperparameters $\alpha$ (used in **Memory Calibration**), $m$ (used in **Reduced Candidate Set**) and $\beta$ (used in **Cycle Consistent Checking**).

**Parameter Study on $\alpha$.** In Memory Calibration (Section 3.2.1), we define feature-wise and position-wise anchors for selecting processed frames whose predicted objects are feature-wise and position-wise accurate, and we define a hyperparameter $\alpha$ to balance these two factors. Larger $\alpha$ means the selection focuses more on the position-wise correctness. As shown in Table 8, (1) when $\alpha = 0$ or $\alpha = 1$, only one of feature-wise and position-wise anchors is considered, and the AUC scores are both 73.7%; (2) when $\alpha = 0.3$, the best performance can be achieved, where the AUC score can be as high as 74.9%, showing the necessity to introduce both anchors, and the effectiveness of our method.

**Parameter Study on $m$.** We experiment with the hyperparameter $m$ defined in Reduced Candidate Set (Section 3.2.1), and present the results in Table 9. We would like to note: (1) using larger $m$, i.e., regarding more processed frames as candidates for memory calibration, would make the inference

Table 8: Experiment on $\alpha$.

| $\alpha$ | LaSOT | | |
|---|---|---|---|
| | AUC | $P_{norm}$ | P |
| 0.0 | 73.7 | 82.0 | 79.9 |
| 0.1 | 74.7 | 83.3 | 81.3 |
| 0.2 | 74.7 | 83.3 | 81.3 |
| 0.3 | **74.9** | **83.4** | **81.4** |
| 0.4 | 74.5 | 83.0 | 81.0 |
| 0.5 | 74.2 | 82.5 | 80.6 |
| 1.0 | 73.7 | 81.9 | 79.7 |

Table 9: Experiment on the number of reduced candidate set $m$.

| $m$ | LaSOT | | |
|---|---|---|---|
| | AUC | $P_{norm}$ | P |
| 20 | 73.9 | 82.3 | 80.3 |
| 30 | **74.9** | **83.4** | **81.4** |
| 40 | 74.7 | 83.2 | 81.2 |
| 50 | 74.4 | 83.0 | 81.0 |
| 60 | 74.1 | 82.5 | 80.6 |

Table 10: Experiment on the threshold $\beta$ of cycle consistent checking.

| $\beta$ | LaSOT | | |
|---|---|---|---|
| | AUC | $P_{norm}$ | P |
| 0.5 | 74.3 | 83.0 | 81.1 |
| 0.6 | 74.8 | 83.4 | 81.4 |
| 0.7 | 74.9 | 83.4 | 81.4 |
| 0.8 | 74.8 | 83.1 | 81.2 |
| 0.9 | 74.7 | 82.8 | 80.8 |

speed slower, and costs more in memory; (2) larger $m$ cannot guarantee better performance, because in videos containing fast-moving objects, the positions of objects change rapidly, and earlier frames might introduce misleading position information; (3) the best performance can be achieved when $m = 30$, appearing to be both effective and efficient.

**Parameter Study on $\beta$.** In Cycle Consistent Checking (Section 3.2.2), we define a threshold $\beta$ to post-check whether the generated positional mask prompts are accurate or noisy. The impacts of different thresholds $\beta$ are illustrated in Table 10, where larger $\beta$ means the prompt would be less frequently used. From the table, we can observe: (1) it is necessary to use a checking mechanism to determine if we can use the generated prompt for the current frame; (2) when $\beta = 0.7$, the trade-offs between positional information and noises can be well made.

# E ADDITIONAL FIGURES

In this section, some additional figures are provided, including the naive version of using SAM2 for VOT (Appendix E.1), the visual impacts of memory calibration (Appendix E.2), more visualizations of positional mask prompt (Appendix E.3), and more qualitative results (Appendix E.4).

## E.1 SAM2 FOR VISUAL OBJECT TRACKING

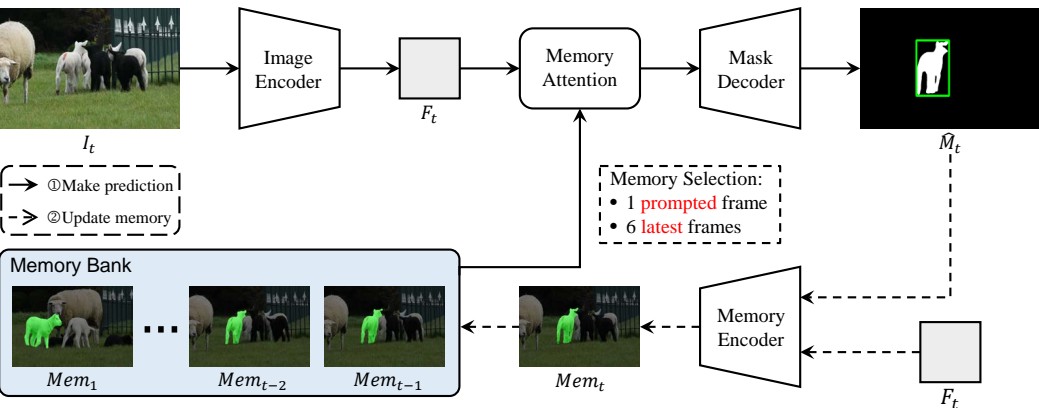

Figure 7: Illustration of using SAM2 for Visual Object Tracking. The first frame $I_1$ is prompted with GT bounding box, and would not be forwarded to Memory Attention for feature conditioning. The mask predictions would be converted to bounding boxes for VOT.

As depicted in Figure 7, SAM2 can be used for VOT by (1) providing the GT bounding box to prompt the first frame, and (2) converting the predicted masks to bounding boxes. The pipeline includes:

**Initialize Tracking with Prompt.** The first frame $I_1$ is forwarded to Image Encoder to extract its features $F_1$, which are decoded by Mask Decoder with GT bounding box prompt. The predicted

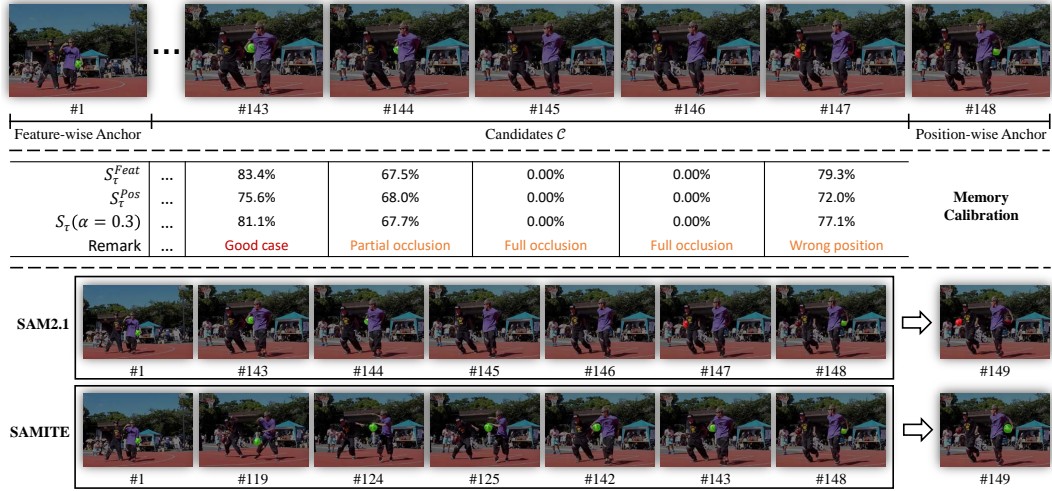

Figure 8: Visual impacts of Memory Calibration. The example is taken from basketball-11 of LaSOT, where the current frame is #149. The default memory selection strategy of SAM2.1 (Ravi et al., 2024) would select the memories of #1 and #143 to #148, regardless of their feature-wise and position-wise correctness, leading to wrong tracking results. Instead, our SAMITE would quantify the feature-wise and position-wise correctness of each processed frame, and select the best ones for conditioning the current frame, ensuring more stable and more accurate tracking.

mask $\hat{M}_1$ and image features $F_1$ are processed by Memory Encoder to obtain the corresponding the encoded memory $Mem_1$, which is stored in Memory Bank.

**Memory-conditioned Tracking.** Any other frames $t$ is uniformly forwarded to Image Encoder to extract features $F_t$. Then, the memories of the first prompted frame and 6 most recent frames are selected to condition the current frame's features as $\hat{F}_t$, via Memory Attention. Next, the enhanced features $\hat{F}_t$ are decoded into mask prediction $\hat{M}_t$. Finally, the memory $Mem_t$ of the current frames would be encoded and added to Memory Bank, used to condition subsequent frames.

### E.2 VISUAL IMPACTS OF MEMORY CALIBRATION

In Figure 8, we provide an example where the original memory selection strategy of SAM2.1 (Ravi et al., 2024) would result in wrong predictions in frame #149, while the designed Memory Calibration can resolve this error by selecting feature-wise and position-wise accurate memories.

Kindly recall that the default memory selection strategy of SAM2.1 is selecting the memories of the first prompted frame and 6 most recent frames, i.e., #1, #143, #144, #145, #146, #147 and #148 in this case. However, as we can observe from the figure, (1) the basketball in #144 is partially occluded, which may provide the model with ambiguous tracking target, e.g., a specific part of the basketball, instead of the complete one; (2) the basketballs in #145 and #146 are fully occluded, which may provide with wrong semantics like nothing needs to be tracked; (3) in #147, two boys just finish behind-the-back dribbles, while the basketball close to the boy in black appears to be easier to be recognized, and the model wrongly considers it as the target, raising a position-wise error. As illustrated in the bottom part of the figure, despite the low quality of the memories of these frames, SAM2.1 would use them to condition frame #149, and wrongly predict the distractor (i.e., the basketball in hand of the boy in black) as the target.

Instead, we propose Memory Calibration to quantify the feature-wise and position-wise errors of each processed frame, and select the best memories to condition the current frame based on the quantified scores. As shown in the middle part of the figure, frame #1 and #148 would be regarded as feature-wise and position-wise anchors, respectively, since the predictions of frame #1 is definitely accurate, owing to the input of GT bounding box, and the object in frame #148 is surely close to that in frame #149. Then, we calculate 2 prototype-wise cosine similarities between the prototypes

of candidates and the anchors. From the figure, we can observe: (1) the cases of occlusions (e.g., frame #144, #145, #146) can be recognized, as their $S_\tau^{Feat}$ scores are lower than others; (2) the position-wise in frame #147 can also be detected, since its $S_\tau^{Pos}$ score is lower than other good cases (e.g., frame #143). After filtering inappropriate frames, our SAMITE finally select frame #1, #119, #124, #125, #142, #143 and #148, and use their memories to condition the current frame #149, intercepting the propagation of position-wise error in frame #147 and achieving correct tracking.

### E.3 MORE VISUALIZATIONS OF POSITIONAL MASK PROMPT

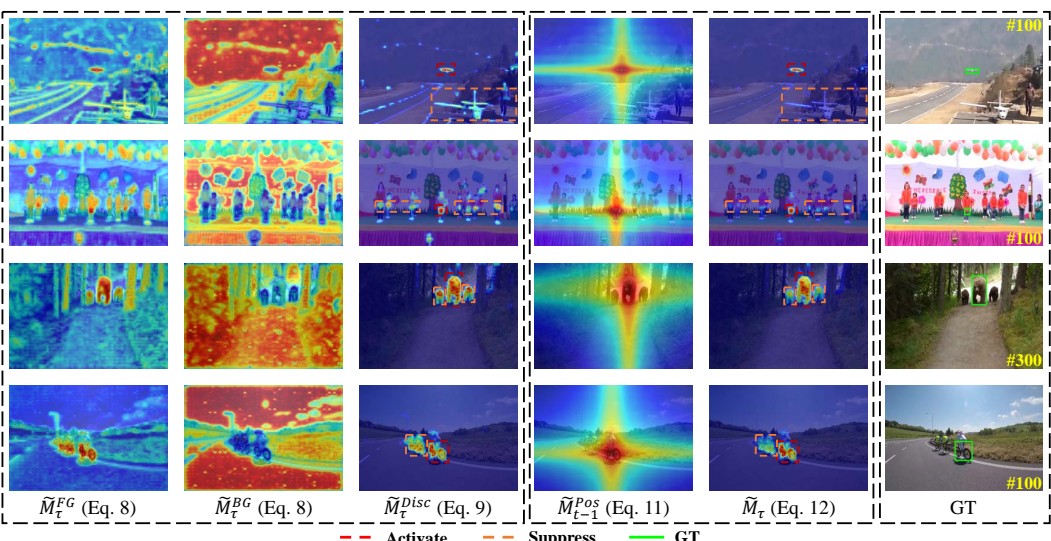

Figure 9: More visualizations of positional mask prompt.

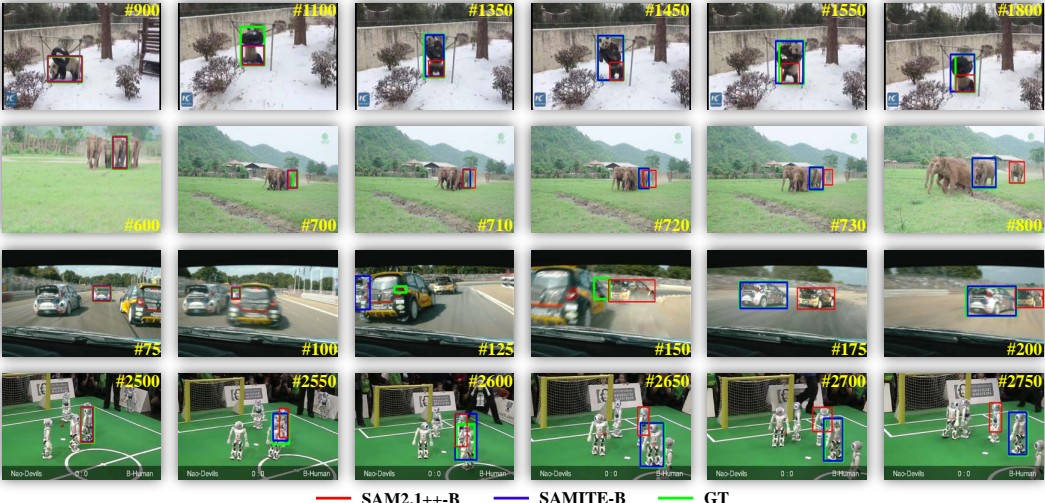

Figure 10: More qualitative results.

In this section, we display more visualizations of positional mask prompt (PPG) in Figure 9, where the first 3 columns refer to essential discriminative prior masks (Xu et al., 2024), and the 4th and 5th columns show positional prior masks and the rectified prior masks (i.e., positional mask prompts). It can be observed from the figure that the generated positional mask prompts can activate the target objects, while suppressing the distractors well. As the model is informed of the positional information about the target objects, it will be less likely to track on wrong objects, better at dealing with

the cases with distractors. One potential drawback of PPG is the generated prompts would never be as accurate as the unavailable GTs, which may introduce some noises to decrease the performance. Therefore, we additionally couple the generated pseudo prompts with a Cycle Consistent Checking mechanism to check the quality of each prompt, please refer to Section 3.2.2 and Appendix D.4 for more details.

### E.4    MORE QUALITATIVE RESULTS

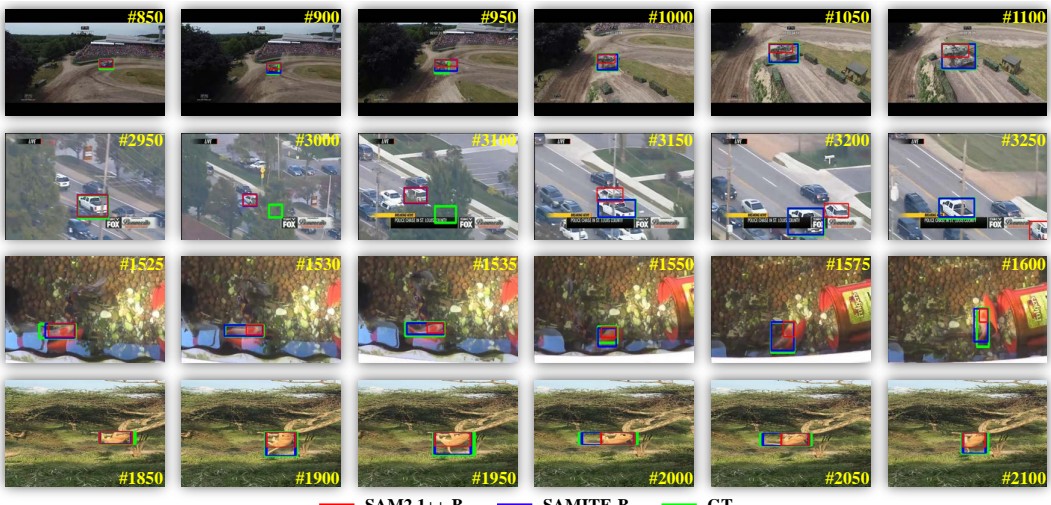

Figure 11: More qualitative results.

More qualitative comparisons between the designed SAMITE and SAM2.1++ (Videnovic et al., 2024) are displayed in Figure 10 and Figure 11, where our SAMITE appears to be better at dealing with occlusions and distractions than SAM2.1++. Particularly, SAMITE is capable of intercepting the propagation of errors. For example, in the 3rd row of Figure 10, when the target car is occluded by the distractor since frame #100, SAMITE (blue rectangle) would then track on the wrong car in #125. With the help of Memory Calibration, SAMITE could correct the error and re-track on the correct car in frame #175. Similarly, in the 4th row of Figure 10 and the 1st row of Figure 11, SAMITE (blue rectangles) makes mistakes on frame #2600 and #950, but can re-focus on the target object in frame #2650 and #1000, respectively, demonstrating the superiority of our module designs.

## F    LIMITATION

Although the proposed zero-shot SAMITE model have already achieved appealing performance on multiple VOT benchmarks, there exist 2 potential limitations or future directions, including: (1) Efficiency: The designed Prototypical Memory Bank (PMB) and Positional Prompt Generator (PPG) merely introduce additional linear computational complexity, while the inference speed (shown in Table 7) appears to be slower than that of the original SAM2.1. Hence, one possible future direction is to reduce some computations in PMB and PPG; (2) In-domain effectiveness: In Table 1 and Table 2, zero-shot SAM2-based methods show their superiority over supervised VOT methods on out-domain testing cases (e.g., LaSOT$_{ext}$ and GOT-10k), the in-domain performance is not that good in datasets like TrackingNet and OTB. We attribute to the fact that these datasets contain some domain-specific knowledge, but it is not captured by SAM2-based methods. Therefore, another possible future direction is to explicitly mine and inject the domain-specific knowledge to SAM2 model.

