# OpenReview forum: "SAMITE: Position Prompted SAM2 with Calibrated Memory for Visual Object Tracking"
_ICLR.cc/2026/Conference — Submitted to ICLR 2026_

### Official Review · Reviewer_E9Se · 2025-10-15

**Soundness:** 3
**Presentation:** 3
**Contribution:** 2
**Rating:** 4
**Confidence:** 4

**Summary:**

This paper introduces SAMITE, a zero-shot Visual Object Tracking (VOT) model built upon SAM2, addressing key challenges in occlusions and distractions. SAMITE has two modules: (1) Prototypical Memory Bank (PMB), which quantifies feature-wise and position-wise correctness of tracking results to select accurate memories, intercepting error propagation; and (2) Positional Prompt Generator (PPG), which generates positional mask prompts to distinguish target objects from distractors. ​ Experiments on six benchmarks demonstrate SAMITE's good performance. ​

**Strengths:**

The following to me are incremental contributions to SAM2:

1. SAMITE identifies and tackles error propagation caused by occlusions and distractions, ensuring more stable and accurate long-term tracking. ​

2. SAMITE achieves strong performance without requiring training on VOT datasets in a zero-shot manner, leveraging the robust video foundation model SAM2. ​

3. The model is tested on six diverse benchmarks, demonstrating excellent performance. SAMITE outperforms state-of-the-art methods in metrics like AUC and precision, and qualitative results show its ability to recover from tracking errors.

**Weaknesses:**

1.  The additional computational complexity introduced by PMB and PPG results in slower inference speeds compared to the original SAM2 model. ​

2. While SAMITE excels in out-domain testing (e.g., LaSOText and GOT-10k), its performance on in-domain datasets like TrackingNet and OTB is less competitive, highlighting a gap in handling domain-specific knowledge.

3. Although codes are submitted in the supplemental materials, I can't find the VOT tracking demos for this VOT papper.

**Questions:**

SAMITE in essence operates behind to two perceptual grouping principles: similarity and proximity with some consideration of common fate. With (over)emphasis on the former two the paper’s complex engineering contributions focus on PMB and PPG resulting in a few points incremental improvement over SOTA. As SAM2 is promptable, just like reasoning segmentation, combining with MLLM may better utilise the third principle to produce more substantial improvement for ICLR 2016. That said,

1. How does SAMITE perform on larger-scale datasets or higher-resolution videos, and are there any scalability challenges?

2. Can we incorporate domain-specific knowledge into SAMITE to improve in-domain performance on datasets like TrackingNet and OTB? ​

3. Could you elaborate on how the hyperparameter α was chosen for balancing feature-wise and position-wise anchors in memory calibration? ​

**Details Of Ethics Concerns:**

No ethics concern.

---

### Official Review · Reviewer_qpGL · 2025-10-16

**Soundness:** 3
**Presentation:** 3
**Contribution:** 2
**Rating:** 4
**Confidence:** 4

**Summary:**

This work proposed **SAMITE**, which is an approach for the Visual Object Tracking task that improves upon existing methods by adapting the **SAM2 video foundation model** and specifically addressing challenges like **occlusions, distractors, and error propagation**. It achieves this through two key additions: the **Prototypical Memory Bank (PMB)** and the **Positional Prompt Generator (PPG)**. The PMB quantifies the **feature-wise and position-wise correctness** of prior tracking results, allowing it to select only the most accurate frames, thereby intercepting the propagation of tracking errors from occluded or distracted objects. Concurrently, the PPG generates **positional mask prompts** to provide explicit location cues for the target, further enhancing tracking accuracy against distractors. By integrating these modules, SAMITE shows superior performance across multiple VOT benchmarks.

**Strengths:**

- Paper is well written and easy to follow, especially Figure 2 illustrate the framework very clearly.
- The idea of decoupling foreground and background feature make sense to me.
- Performance is good compared to existing SAM2 variants across multiple benchmarks.

**Weaknesses:**

- The method should be applicable to the VOS task, not just VOT. The work should include more results on VOS benchmarks to demonstrate its generalizability.

- The zero-shot claim for SAMITE is not very convincing. Since SAM2 itself is trained on a large-scale VOS dataset, adapting it to the simpler VOT task should not be considered zero-shot.

- The proposed method appears to maintain a better memory bank due to the designed memory selection strategy. However, I believe there is still a significant gap from "preventing error propagation," especially since the method heavily relies on recent memory and a reduced candidate set in the temporal domain. Under high FPS settings, a single tracking error can quickly fill the recent memory with features from the wrong object, triggering error propagation. Therefore, the claim of error propagation interception seems overclaimed.

- Table 3 shows that applying PPM without the CCC causes performance to drop. This indicates that the prompt generation mechanism is inherently unreliable and potentially noise-introducing, requiring a subsequent CCC to selectively filter its own output.

**Questions:**

I hope the authors can address my concerns in weakness, especially:
- More results on VOS benchmark to show effectivness of SAMITE. I do not see any reason why it can not be applied to VOS task.
- The zero-shot claim, perhaps training-free is a better term.
- The claim in preventing error propagation.

---

My current rating is 4 (boarderline reject) but more close to 5 (boarderline). I am willing to adjust my rating according to other reviewers' comments and the author response.

---

### Official Review · Reviewer_e8gc · 2025-10-25

**Soundness:** 3
**Presentation:** 3
**Contribution:** 2
**Rating:** 4
**Confidence:** 4

**Summary:**

The paper proposes SAMITE, a zero-shot visual object tracker built on SAM2. It introduces a Prototypical Memory Bank (PMB) to select reliable memory frames using similarity to anchor frames, and a Positional Prompt Generator (PPG) to refine prior masks with positional cues and a cycle-consistency check. These modules aim to reduce error propagation from occlusions and distractors. Experiments on several benchmarks show that SAMITE outperforms SAM2-based baselines in tracking accuracy and robustness.

**Strengths:**

1. Clear motivation addressing occlusion and distraction failure modes and visual examples.

2. Achieve state-of-the-art performance across six standard VOT benchmarks with several metrics.

3. Good ablation study (Table 3) shows the additive contribution of proposed modules (PMB, PPG, CCC).

**Weaknesses:**

1. Limited novelty: The proposed components, such as the prototypical memory bank and prior-guided mask generation, are largely extensions or combinations of existing ideas from prior works (e.g., SAM2-based memory selection, AENet prior masks), offering only incremental technical contributions.

2. Missing runtime analysis: The paper does not report runtime or computational overhead compared to SAM2 or related trackers, leaving efficiency and practicality unclear.

3. Missing qualitative results: No video demos are provided in the supplementary materials. Considering the main application is visual object tracking, it is necessary to provide video demos in order to access the temporal smoothness of tracking results.

**Questions:**

Please address my comments in paper weaknesses.

---

### Official Review · Reviewer_Ke83 · 2025-10-31

**Soundness:** 2
**Presentation:** 2
**Contribution:** 2
**Rating:** 2
**Confidence:** 5

**Summary:**

The paper proposes SAMITE, a plug-and-play enhancement for SAM2-based video object tracking that reduces error propagation from bad memory frames. It does this with (1) a Prototypical Memory Bank that keeps only frames most similar to ground-truth and recent targets, and (2) a Positional Prompt Generator that builds location-aware mask prompts so the tracker can re-localize the target during occlusions or distractions, yielding small but consistent gains on six VOT benchmarks over SAM2.1++ and other zero-shot trackers. However, the proposed solution is a heuristic extension of existing memory-selection ideas, lacks deeper analysis (especially on efficiency and robustness), and the presentation leaves core steps underspecified. For a top-tier conference, the contribution is incremental rather than fundamental.

**Strengths:**

1.	The paper pinpoints a real challenge in SAM2-style, memory-based VOT. Once a bad frame is written to memory (due to occlusion or a look-alike distractor), the error keeps getting reused.
2.	The paper proposes two modules, the Prototypical Memory Bank (PMB) and Positional Prompt Generator (PPG). These two modules are conceptually simple and slot on top of SAM2 without retraining to tackle the problem of error propagation
3.	The authors do report results on six standard VOT benchmarks (LaSOT, LaSOText, GOT-10k, TrackingNet, NFS, and OTB), and they compare to both supervised trackers and SAM2-based models.
4.	The authors explicitly write a limitation section acknowledging efficiency and in-domain gaps, which is good scholarship.

**Weaknesses:**

1.	They themselves say inference is slower than SAM2.1 because PMB/PPG add work. For a tracker, that’s serious. They have 9.2 FPS with a base backbone with 78.9 AO in GOT-10k, which is less than SAMURAI by 0.7% which has a higher FPS of 14.3 FPS.
2.	SAMURI is not shown in Table 2, which probably indicate that its performance is better than SAMITE. If this is the case, this means that SAMURI outperforms SAMITE in 4 datasets out of 6. As shown in the paper, all ablation studies are done on LaSOT and LaSOText, which are the only two datasets that SAMITE excels in.
3.	 Figure 4 (qualitative) is cherry-picked. They pick cases where SAM2.1++ blatantly fails and SAMITE recovers. No counterexamples.
4.	They “empirically set α=0.3” later, but they don’t justify why the feature-wise anchor should dominate the position-wise anchor. Since they argue occlusions are common, a nearer frame (position-wise) might actually be worse than the GT frame. This should have been validated.
5.	 They tell us PPG sometimes hurts, and then they add the Cycle-Consistent Checking (CCC) to gate it (Table 3: 71.9 → 73.2). But they never show how often CCC actually suppresses PPG.
6.	  Using cosine over pooled features to decide which memories to keep is a common pattern in VOS/VOT memory networks. Here it’s just applied twice (feature-wise, position-wise). That’s evolutionary, not revolutionary.
7.	 Everything is done zero-shot on top of a frozen SAM2. That’s nice for practicality, but for a top-tier conference, we often expect either a learned selection policy, or a theoretical justification, or at least an analysis of failure modes as a function of similarity thresholds. None is here.
8.	 Since the whole pitch is “we intercept error propagation,” I’d expect a test where they force a bad memory at frame t and show whether SAMITE recovers faster than SAM2.1++ on the same video.
9.	 Impact is tied to SAM2 staying dominant. The method is tightly coupled to the SAM2 memory format and promptable decoder; it is not shown to work on non-SAM2 video foundation models.
10.	The claim of being “the first to identify error propagation” is not properly justified, given that SAMURAI literally has an occlusion likelihood to filter memories and SAM2.1++ has a distractor-aware memory. That should be toned down.

**Questions:**

see above weaknesses

---

### Meta-Review · Area_Chair_YPqe · 2026-01-06

**Summary:**

In the initial phase, all reviewers gave negative scores (2, 4, 4, 4). They raised concerns from different aspects.

Reviewer Ke83: questioned the method's efficiency and performance, and highlighted unjustified claims.

Reviewer e8gc: raised issues regarding novelty, the absence of runtime analysis, and the lack of qualitative results.

Reviewer qpGL: pointed to insufficient experiments, unjustified claims, and concerns about the robustness of the proposed method.

Reviewer E9Se: expressed concerns about efficiency and performance.

**Reviewer Concerns:**

Since no rebuttal is provided, all concerns remain unsolved.

**Reviewer Scores:**

Since no rebuttal is provided, the scores will remain unchanged.

---

### Decision · Program_Chairs · 2026-01-26

Reject